# Toward Robust Hyper-Detailed Image Captioning: A Multiagent Approach and Dual Evaluation Metrics for Factuality and Coverage

**Saehyung Lee** [1] [*]  **Seunghyun Yoon** [2]  **Trung Bui** [2]  **Jing Shi** [2]  **Sungroh Yoon** [1] [3]

## Abstract

Multimodal large language models (MLLMs) excel at generating highly detailed captions but often produce hallucinations. Our analysis reveals that existing hallucination detection methods struggle with detailed captions. We attribute this to the increasing reliance of MLLMs on their generated text, rather than the input image, as the sequence length grows. To address this issue, we propose a multiagent approach that leverages LLM-MLLM collaboration to correct given captions. Additionally, we introduce an evaluation framework and a benchmark dataset to facilitate the systematic analysis of detailed captions. Our experiments demonstrate that our proposed evaluation method better aligns with human judgments of factuality than existing metrics and that existing approaches to improve the MLLM factuality may fall short in hyper-detailed image captioning tasks. In contrast, our proposed method significantly enhances the factual accuracy of captions, even improving those generated by GPT-4V. Finally, we highlight a limitation of VQA-centric benchmarking by demonstrating that an MLLM's performance on VQA benchmarks may not correlate with its ability to generate detailed image captions. Our code and data are available at https://github.com/adobe-research/CapMAS.

## 1. Introduction

Numerous image captioning methods utilizing deep neural networks (DNNs) have been proposed (Vinyals et al., 2015; Xu et al., 2015). However, they are generally limited to generating short captions, which constrains their broader application in real-world scenarios. For instance, in cases such as assistance for visually impaired individuals, where it is necessary to provide highly detailed descriptions of the scene in front of the user, these methods may not be suitable.

Following the recent success of large language models (LLMs) (Brown et al., 2020), there have been attempts to use text and information from other modalities as input to LLMs. Notably, many studies have explored multimodal large language models (MLLMs) that incorporate visual information (Li et al., 2023a; Dai et al., 2023; Liu et al., 2024b). These models have demonstrated significantly superior performance compared to traditional models in tasks such as visual question answering (VQA) and captioning (Liu et al., 2024a). In particular, MLLMs, leveraging the advanced language capabilities of LLMs, are able to generate much longer and more detailed captions than conventional captioning models. However, these generated captions frequently contain inaccurate information, including descriptions of objects that are not present in the input image (Leng et al., 2024). Such hallucination problems hinder the practical application of MLLMs in real-world settings.

Three major approaches have been recently proposed to improve the factuality of MLLMs: (i) Decoding-based methods (Leng et al., 2024) reduce the probabilities of hallucination-related tokens during the model's decoding process without requiring additional training; (ii) Training-based methods (Liu et al., 2023a) further train the models on curated multimodal datasets to ensure they generate only accurate responses; (iii) Corrector-based methods (Zhou et al., 2024) employ a corrector model that detects and either removes or revises hallucinations present in the model's responses.

In this paper, **we propose Caption factuality enhancing MultiAgent System** (CapMAS), a multiagent approach to correct hyper-detailed image captions. Unlike existing corrector-based approaches that require training a corrector (Lee et al., 2024), CapMAS improves the factuality of detailed image captions by leveraging the collaboration between an LLM and MLLM, without the need for additional training. Moreover, unlike methods that target specific types of hallucinations (Li et al., 2023b; Zhou et al., 2024), our approach does not pre-define the hallucination types, allowing

---

[*]Work done during internship at Adobe Research. [1]Department of Electrical and Computer Engineering, Seoul National University [2]Adobe Research [3]Interdisciplinary Program in Artificial Intelligence, Seoul National University. Correspondence to: Sungroh Yoon <sryoon@snu.ac.kr>.

*Proceedings of the $42^{nd}$ International Conference on Machine Learning*, Vancouver, Canada. PMLR 267, 2025. Copyright 2025 by the author(s).

it to address a broader range of issues. The method proceeds as follows: (i) an LLM decomposes a given detailed caption into atomic propositions; (ii) an MLLM verifies the truthfulness of each atomic proposition based on the image; and (iii) the LLM revises the caption accordingly. Our design is particularly motivated by the observation that, as the length of a model's response increases, hallucinations generated later in the sequence become more difficult for existing methods (Wang et al., 2023; Zhou et al., 2024) to detect.

Evaluating the factuality of detailed captions is not straightforward. Through experiments, we demonstrate that conventional caption evaluation metrics such as BLEU (Papineni et al., 2002), ROUGE (Lin, 2004), METEOR (Banerjee & Lavie, 2005), and CIDEr (Vedantam et al., 2015), as well as recently proposed methods (Hessel et al., 2021; Petryk et al., 2024), fail to accurately assess the factuality of detailed captions. To address this issue, **we propose a GPT-based method for factuality evaluation and validate its effectiveness through experiments that include human evaluations**. Even if a caption contains factual information, however, it may still be considered inadequate if it does not sufficiently capture the visual information. To measure the coverage of captions, **we construct a detailed VQA dataset through a collaboration between humans and an AI agent** (Achiam et al., 2023). If a caption fully encapsulates the information of a given image, questions about the image should be answerable accurately using only the caption, without referencing the image itself.

Our experiments surprisingly reveal that methods designed to improve the factuality of MLLMs, which have proven effective in tasks like VQA (Huang et al., 2024), may be ineffective for hyper-detailed image captioning tasks that require longer responses. In contrast, CapMAS significantly enhances the factuality of captions and can be applied in a plug-and-play manner to any captioning model; notably, this improvement extends to captions generated by the state-of-the-art closed model, GPT-4V (Achiam et al., 2023). Finally, we highlight an issue with the current VQA-centric benchmarking (Duan et al., 2024) by showing that an MLLM's performance on VQA benchmarks may not correlate with its ability to generate detailed image captions.

## 2. Related Work

**Multimodal large language models.** LLMs that process inputs from multiple modalities, including text and other types of data, are referred to as multimodal LLMs (Yin et al., 2023a). Among these, **LLMs that handle visual input have been the most actively researched, and the MLLMs discussed in this paper are focused on this category**. Research on these models primarily explores methods for fusing the output of an independent vision encoder into the input of an LLM. The BLIP models (Li et al., 2023a;

Dai et al., 2023) align the frozen vision encoder and LLM using a lightweight transformer (Vaswani, 2017) called Q-Former. The trainable input tokens of the Q-Former interact with the output tokens from the vision encoder through cross-attention, transforming them into input tokens for the LLM. The LLaVA models (Liu et al., 2024b;a) use a simple MLP connector to align the vision encoder with the LLM. All output tokens from the vision encoder, passed through the MLP connector, are used as input to the LLM. The vision encoder's parameters remain fixed during the training of the MLP connector and the LLM. Unlike existing MLLMs, the InternVL models (Chen et al., 2024c;b) have demonstrated the effectiveness of increasing the size of both the vision encoder and the vision-language connector. They utilize a 6-billion parameter vision encoder and an 8-billion parameter vision-language connector. The connector is obtained by fine-tuning the pre-trained multilingual LLaMA (Cui et al., 2023). Despite the many advancements in open-source MLLMs, closed-source MLLMs such as GPT-4V or GPT-4o[1] still outperform them significantly. As a result, these GPT models represent the upper bound performance in benchmarks and are commonly used to evaluate MLLMs (Petryk et al., 2024). In our work, we demonstrate that captions generated by GPT-4V can be improved using our method, and we use GPT-4o to evaluate captions.

**MLLM hallucinations and mitigation strategies.** MLLMs sometimes generate inaccurate responses. For example, they may incorrectly describe the characteristics of objects in an input image, misrepresent relationships between objects, or even describe objects that do not exist. To mitigate these hallucination problems, decoding-based methods apply penalties to the probabilities of tokens that are likely to be hallucinations during the decoding process. For instance, VCD (Leng et al., 2024) induces hallucinations using corrupted images, while OPERA (Huang et al., 2024) leverages the correlation between high attention weights assigned to a few summary tokens and hallucinations. Training-based methods focus on exploring training data that can suppress the generation of hallucinations. Liu et al. (2023a) demonstrated that hallucinations can be alleviated by incorporating negative samples—descriptions that explicitly state the absence of certain objects in a given image—into visual instruction tuning datasets. Corrector-based methods (Zhou et al., 2024; Lee et al., 2024) detect, remove, and revise hallucinations present in MLLM responses by using a corrector model. This model is obtained by supervised fine-tuning a pre-trained MLLM. The corrector model then revises the initial response based on the given image.

---

[1]https://openai.com/index/hello-gpt-4o/

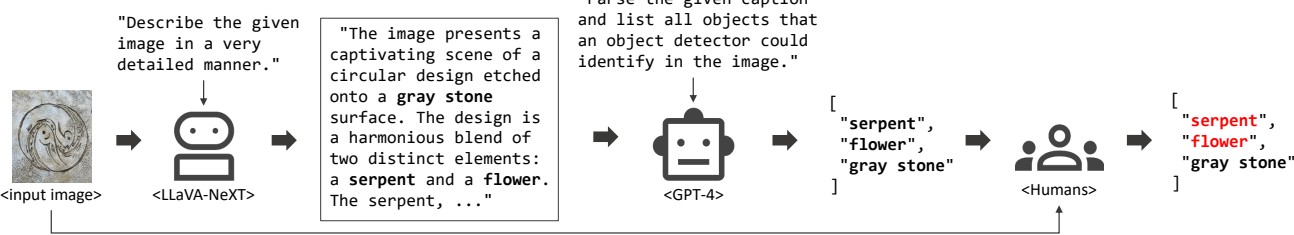

Figure 1: The process of generating a data sample for evaluating hallucination detection methods in detailed image captioning tasks. Human annotators identify and label object hallucinations within the caption generated by LLaVA-NeXT (Liu et al., 2024a) for an image.

**Caption evaluation methods.** Since short image captions are relatively easy to obtain reference captions for, we can use matching-based caption evaluation methods (Hossain et al., 2019) to assess them. However, for long and detailed captions generated by MLLMs, the number of reference captions required for such evaluations becomes exceedingly large. Thus, it becomes impractical to evaluate detailed captioning using traditional approaches. Hessel et al. (2021) proposed CLIPScore, a reference-free evaluation method. CLIPScore measures the distance between an image and its caption within the joint representation space of CLIP (Radford et al., 2021). Additionally, the authors introduced RefCLIPScore, which uses both the image and reference captions within that same representation space. Chan et al. (2023) addressed the limitations of matching-based methods by utilizing an LLM. The LLM-based metric they proposed, CLAIR, assigns scores to captions based on reference captions using an LLM. Similarly, ALOHa (Petryk et al., 2024) detects object hallucinations by comparing generated captions to reference captions using an LLM.

## 3. Method

In this paper, we propose a multiagent-based caption correction method. Corrector-based methods typically detect and remove hallucinations within model responses. Unlike existing approaches, which require the corrector model training, our method employs collaboration between an MLLM and LLM. Moreover, in contrast to previous methods that are limited to correcting specific types of hallucinations (Zhou et al., 2024), our approach is free from such constraints. We also propose a framework for evaluating the detailed image captioning capabilities of an MLLM. Unlike existing methods, our proposed evaluation approach allows for assessing image captioning models in terms of both factuality and coverage, evaluating each of these aspects separately.

### 3.1. Motivating Observations

Here, we examine the performance of existing hallucination detection methods on tasks that require generating long responses. To facilitate these analyses, we construct a dataset

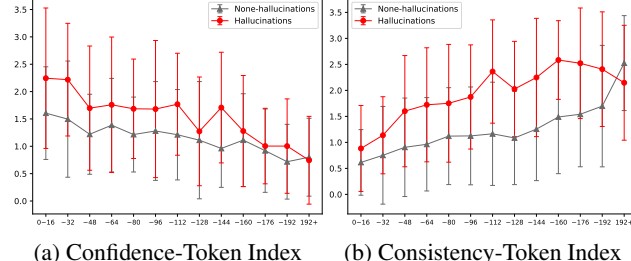

(a) Confidence-Token Index    (b) Consistency-Token Index

Figure 2: The hallucination scores of the Confidence and Consistency methods based on object positions within detailed captions. Object hallucinations near the end of the captions (192+) are undetectable by both methods.

as follows: (i) We prompt an MLLM with "Describe the given image in a very detailed manner." and collect the model's responses for a specified image set; (ii) For the convenience of our analysis, we use an LLM to identify objects that may be hallucinations; (iii) Human annotators then label each parsed object as either a hallucination or not, based on the corresponding image. We use LLaVA-NeXT (Liu et al., 2024a) and GPT-4 as the MLLM and LLM, respectively. Figure 1 illustrates the process of constructing the dataset. To build the dataset, we use a subset of IIW-400 (Garg et al., 2024). We detect object hallucinations using two of the most widely adopted hallucination detection methods:

1. **Confidence** (Zhang et al., 2023; Zhou et al., 2024): This method detects hallucinations using the predicted probability $p_{obj}$ for the object token during LLaVA-NeXT generation. For multi-token objects, the token probabilities are multiplied. The hallucination score $H_{obj} = -\log p_{obj}$, increases with the likelihood of hallucination.

2. **Consistency** (Wang et al., 2023; Zhao et al., 2024): This method assumes that hallucinations are sensitive to decoding randomness. Using stochastic decoding, we have LLaVA-NeXT generate 40 captions per image and count the occurrence $t_{obj}$ of each object in the dataset of Figure 1. The hallucination score is $H_{obj} = -\log \frac{t_{obj}}{40}$.

Figure 2 presents the hallucination scores of each method by the position of objects appearing within detailed cap-

Table 1: Performance comparison of hallucination detection methods for the dataset of Figure 1.

| Method | AUROC↑ | FPR95↓ |
|---|---|---|
| Confidence | 57.5 | 95.1 |
| Consistency | 73.5 | 75.6 |
| Object Detector | 61.5 | 95.7 |
| Isolation | **81.4** | **71.7** |

tions. The horizontal axis of the graphs represents bins of object token indices, with larger token indices indicating positions closer to the end of the caption. The vertical axis represents the mean and standard deviation of the hallucination scores within each bin. Note that Figure 2a reflects the positions and hallucination scores during greedy decoding, while Figure 2b is derived from the average positions and hallucination scores across 40 stochastic decoding iterations. Figure 2 demonstrates that hallucinations generated after the 192nd token are undetectable by the Confidence and Consistency methods. Based on these results, we can infer that existing hallucination detection methods may be ineffective in detecting hallucinations in long detailed captions.

Our hypothesis regarding these results is that *as MLLM outputs become longer, they become more strongly grounded in the text they generate rather than the given image.* In fact, our hypothesis is supported by several recent studies. For example, Liu et al. (2024c) demonstrated that as MLLM responses lengthen, the attention weights assigned to image tokens decrease, and Zhong et al. (2024) showed that MLLM responses are significantly influenced by prior dialogue. Based on this hypothesis, we test a method for determining whether each object is a hallucination by disconnecting it from its context (**Isolation**). The Isolation method involves querying the LLaVA-NeXT model with parsed objects using the prompt template, "Is there a {} in the photo?" along with the image. When the probability of the "Yes" token for the object query is $p_{\text{Yes}|\text{obj}}$, the hallucination score is defined as $H_{\text{obj}} = -\log p_{\text{Yes}|\text{obj}}$. We compare the object hallucination detection performance of the Isolation method with that of the Confidence method, the Consistency method, and a method based on an object detector (**Object Detector**) introduced in recent studies (Yin et al., 2023c; Ge et al., 2024). We measure their detection performance on the dataset of Figure 1 using Area Under the Receiver Operating Characteristic (AUROC) and False Positive Rate at 95% true positive rate (FPR95). Table 1 demonstrates that the Isolation method outperforms the others. This suggests that breaking a long caption into smaller units and examining each individually can help detect hallucinations in detailed captions.

**Comparison with existing studies.** Actually, the concept of decomposing text into smaller units and assessing the factuality of each has been introduced in prior studies (Min et al., 2023; Jing et al., 2024). Here, we summarize the key differences between our study and previous research:

1. Unlike existing studies, which do not rigorously justify the need for decomposition, we empirically demonstrate the motivation behind our approach (Section 3.1).
2. While previous studies focused solely on proposing evaluation metrics, our research advances further by introducing a system that leverages the decomposition process to generate improved image captions (Section 3.2).
3. Existing evaluation metrics assess factuality using unimodal data (either text or image). In contrast, our proposed evaluation metric utilizes multimodal data (both text and image) for factuality assessment (Section 3.3).
4. Previous studies focus solely on measuring the factuality of generated text. In contrast, our study proposes a method that assesses both the factuality and coverage of any given image caption (Section 3.3).
5. We demonstrate that our metric correlates more strongly with human evaluations than existing metrics and is robust against their critical limitations (Section 4.2).

### 3.2. Caption Factuality Enhancing MultiAgent System

To address various types of hallucinations comprehensively, we first decompose each detailed caption into atomic propositions using an LLM. An atomic proposition is a claim or statement that must either be true or false. For example, the caption "A house has a red roof and a chimney" is broken down into "A house has a red roof" and "A house has a chimney." We use an LLM to perform this process, but we allow flexibility in cases where the results do not strictly conform to the definition of an atomic proposition. We then investigate the truth of each decomposed unit using an MLLM. Each unit is converted into a True/False question and independently fed to the MLLM. The hallucination score $H(u)$ for the unit $u$ is defined as follows:

$$-\log\left(\min\left(p\left(\text{T}|x, Q(u)\right) - p\left(\text{F}|x, Q(u)\right), \epsilon\right)\right) \quad (1)$$

$p\left(\text{T}\right)$ and $p\left(\text{F}\right)$ represent the MLLM's token probabilities for the "True" and "False" tokens, respectively. $x$ and $\epsilon$ denote the input image and a very small constant near zero. $Q(\cdot)$ is a function that converts the input text into a True/False question, which we implement by prepending "True or False?" to the input. Each unit is included in either the True set $\mathcal{T}$ or the False set $\mathcal{F}$, based on its hallucination score. To achieve this, we introduce a hyperparameter $\pi$, such that $\mathcal{T} = \{u|H(u) \leq \pi\}$ and $\mathcal{F} = \{u|H(u) > \pi\}$. Finally, the initial caption, along with the corresponding sets $\mathcal{T}$ and $\mathcal{F}$, is provided to an LLM, which corrects the initial caption to ensure it contains only factual information.

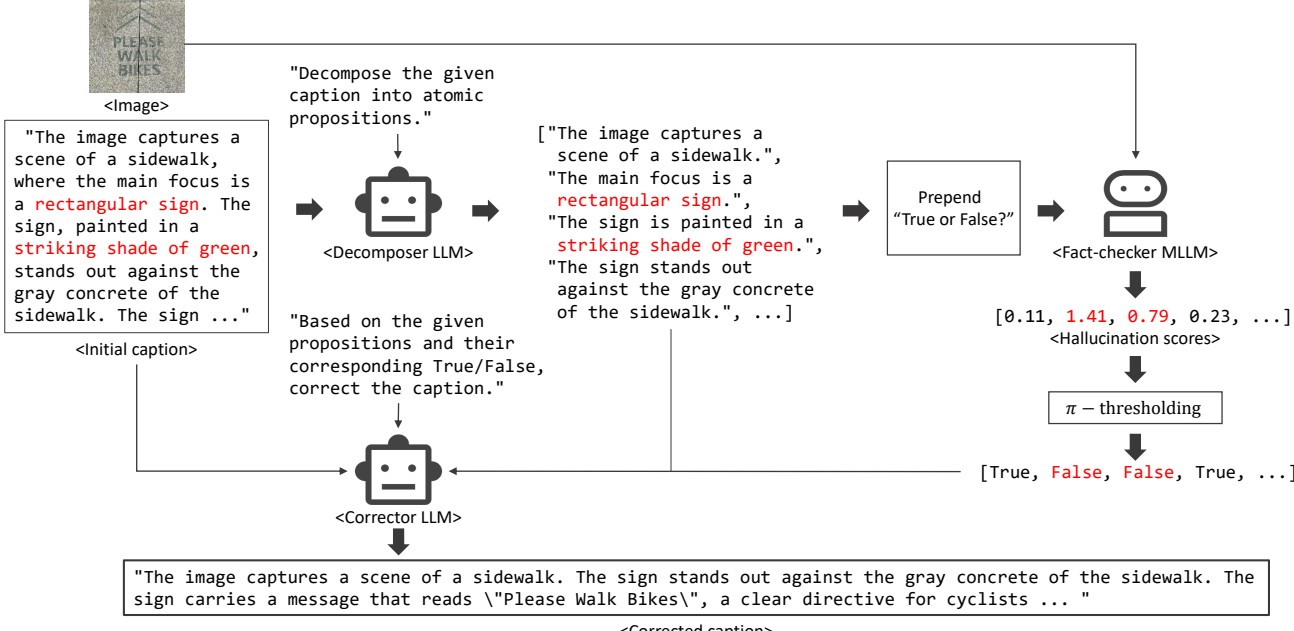

Figure 3: Overview of CapMAS. The decomposer LLM breaks an initial caption into atomic units. These units are converted into True/False questions and fed into the MLLM along with the image, where each unit is assigned a hallucination score according to Equation (1). Units are classified as True or False based on the threshold $\pi$, and the corrector LLM then revises the initial caption accordingly.

We name this pipeline, which improves the factuality of detailed image captions through the collaboration of a pretrained LLM and MLLM, **Caption factuality enhancing MultiAgent System (CapMAS)**. CapMAS is training-free and can be applied in a plug-and-play manner to any captioning model. Unlike existing methods that can only address predefined types of hallucinations, CapMAS can detect and correct all hallucinations at the atomic unit level. The pipeline of CapMAS is illustrated in Figure 3.

### 3.3. Evaluation Methods

Traditional caption evaluation methods rely on word matching with reference captions, suitable for short captions generated by conventional models. However, MLLMs produce longer and more detailed captions, making it impractical to obtain sufficient reference captions for accurate evaluation. Given the enriched content of these image captions, rather than simply evaluating them as good or bad, we aim to assess them systematically by considering two key perspectives:

- **Factuality**: The degree to which the content of the caption is factual and free from hallucinations.
- **Coverage**: The extent to which the caption captures the information contained in the image.

We propose evaluation methods for detailed image captions from these two perspectives.

Table 2: Meta-evaluation results across various caption evaluation methods. DOCCI and its synthetic hallucinatory captions are used for the meta-evaluation. The highest-rated caption for each method is highlighted in **bold**. The full table is in Appendix D.

| Caption | Evaluation Metric | | | | | |
|---|---|---|---|---|---|---|
| | CIDEr | CLIP-S | RefCLIP-S | CLAIR | ALOHa | Ours |
| Clean | 6.4 | 81.3 | 75.5 | **86.9** | 36.2 | **62.8** |
| Object | 4.8 | 81.0 | 75.3 | 85.2 | 31.5 | 52.3 |
| Attribution | 6.2 | 80.9 | 75.2 | 80.0 | 34.3 | 60.9 |
| Relation | **6.7** | **81.4** | **75.6** | 83.5 | **36.9** | 51.9 |

**Factuality.** If a human were to measure the factuality of a text, it would be natural to decompose the text into units that can be classified as true or false, and then calculate the proportion of true units (Maynez et al., 2020). We adopt this approach to measure the factuality of captions, utilizing the state-of-the-art model GPT-4o. In our framework, GPT-4o decomposes each caption into atomic propositions and determines their truthfulness based on the corresponding image and reference caption. If the number of atomic propositions judged as true and false are $T$ and $F$, respectively, the factuality of the caption is defined as $\frac{T}{T+F}$.

To validate this evaluation method, we use the DOCCI dataset (Onoe et al., 2024), which contains human-annotated detailed image captions. Specifically, for each image in a subset of the dataset, we prepare the following four types of

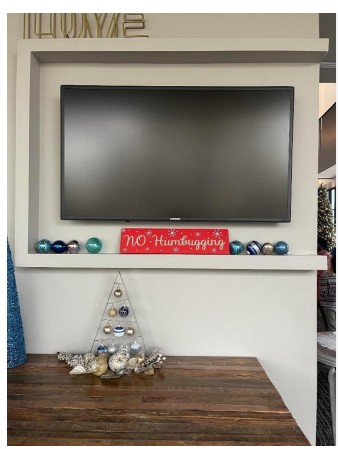

```
1.  What is the main focus of the photo?
    A) A landscape B) A television and decorations C) A group of people D) A building
2.  Where was the photo likely taken?
    A) In a park B) Inside a house C) At a beach D) In a museum
3.  What is the situation depicted in the photo?
    A) A family gathering B) A decorated living space C) A work meeting D) A sporting event
4.  What is in the center of the photo?
    A) A painting B) A television C) A person D) A window

...

47. What is on the left side of the table in the photo?
    A) A lamp B) A blue decorative tree C) A vase D) A stack of books
48. What is the texture of the table in the foreground?
    A) Smooth and shiny B) Rough and rustic C) Soft and cushioned D) Metallic and cold
49. What is in the background of the photo, to the right side?
    A) A kitchen B) A Christmas tree C) A bookshelf D) A window with curtains
50. What type of ornaments are on the triangular decoration on the table?
    A) Animal figurines B) Christmas baubles C) Miniature houses D) Candles
```

Figure 4: An example of our coverage evaluation data sample. The dataset consists of multiple-choice questions with four or fewer options. As demonstrated, the dataset includes questions with varying levels of granularity, ranging from broad to highly detailed. We have an LLM solve these problems using only the provided captions.

captions (details provided in Appendix G):

1. Clean: The original caption (*e.g.*, *An indoor view captures a cat on a wooden floor, attempting to catch a large pale peacock feather flying above it*).
2. Object: A description of an object likely present but not in the image is added to the Clean caption (*e.g.*, *An indoor view captures a cat on a wooden floor, attempting to catch a large pale peacock feather flying above it. A small red ball is rolling near the cat*).
3. Attribution: Some object attributions in the Clean caption are modified to be inconsistent with the image (*e.g.*, *An indoor view captures a cat on a metal floor, attempting to catch a small dark peacock feather flying above it*).
4. Relation: Some object relationships in the Clean caption are altered to conflict with the image (*e.g.*, *An indoor view captures a cat on a wooden floor, attempting to catch a large pale peacock feather flying below it*).

We evaluate the four types of captions using various metrics (BLEU, ROUGE, METEOR, CIDEr, CLIP-S, RefCLIP-S, CLAIR, and ALOHa), including our own, to determine whether the hallucinations in the three modified types are reflected in the scores. For a fair comparison, all methods requiring GPT (CLAIR, ALOHa, and ours) use GPT-4o, and all methods requiring reference captions (BLEU, ROUGE, METEOR, CIDEr, CLAIR, ALOHa, and ours) use a separate set (Garg et al., 2024) of human-annotated captions.

Table 2 shows that existing metrics are unreliable for evaluating the factuality of detailed image captions. Specifically, CLIP can only process up to 77 tokens and operates like a bag-of-words model (Yuksekgonul et al., 2023). This prevents CLIP-based metrics from capturing the full content of detailed image captions, particularly missing Relation hallucinations. ALOHa effectively addresses Object and

Attribution hallucinations but fails to capture Relation hallucinations due to its algorithmic limitations. CLAIR detects and reflects all three types of hallucinations in the scores. However, CLAIR does not focus solely on factuality; instead, it allows the GPT model to directly score each caption, applying the evaluation criteria implicitly defined by the GPT model. In contrast, our metric exclusively considers the factuality of the caption. While it does not assign a perfect score to the Clean captions due to GPT-4o's limitations in image understanding, it successfully assigns the highest score to Clean among the four caption sets.

**Coverage.** An image caption with only factual information is not high-quality if it focuses solely on trivial aspects of the image. To assess the coverage of captioning models, we propose a QA-based metric and a benchmark dataset. Our coverage evaluation method is based on the assumption that *if an image caption fully captures the information in the image, visual questions about that image should be answerable by referencing the caption alone*.

Our goal is to evaluate hyper-detailed image captions. Therefore, the visual questions for evaluation must include a variety of detailed and nuanced questions about the images. Given the limitations of existing VQA datasets in this regard (Yin et al., 2023b; Li et al., 2023b; Yue et al., 2024), we construct a new VQA dataset. However, creating a new VQA dataset that includes a variety of detailed questions requires substantial labor. To reduce the associated costs, we follow the process outlined below to construct our dataset:

1. Generating more than 50 questions per image in the IIW-400 dataset using GPT-4o.
2. Deduplicating the questions for each image using Sentence-BERT (Reimers & Gurevych, 2019).

3. Human labelers refine or remove ambiguous, flawed, or common-knowledge questions.
4. Human labelers annotate the correct answers to the remaining and revised questions.

Our coverage evaluation dataset contains a total of 19,899 multiple-choice questions, with each image averaging 49.8 questions. We present an example of our dataset in Figure 4. While our benchmark dataset can also be used to assess the visual understanding capabilities of MLLMs, we use it to evaluate the coverage of captioning models by having an LLM answer the questions based on the generated captions.

## 4. Experimental Results and Discussion

### 4.1. Experimental Setup

We adopt LLaVA-v1.5-7B, LLaVA-NeXT-7B, LLaVA-NeXT-13B, InternVL-Chat-V1.5, and GPT-4V as the models for both captioning and CapMAS's fact-checking. We use LLaMA-3-8B (AI@Meta, 2024) or GPT-4 as the decomposer and corrector LLMs in CapMAS. Our experiments utilize the IIW-400 and DOCCI datasets, which contain images paired with highly detailed, hallucination-free captions. These high-quality reference captions enable precise evaluation of the captioning models.

We employ our proposed factuality and coverage metrics, along with CLAIR, all based on GPT-4o, to evaluate the generated captions. To ensure robust evaluation, we summarize the captions (Ge et al., 2024) generated from five different prompts using LLaMA-3-8B. The only CapMAS hyperparameter, $\pi$, is tuned on a validation set of five examples sampled from the DCI dataset (Urbanek et al., 2024). The prompt templates are provided in Appendix G.

### 4.2. Our Metric's Correlation with Human Evaluation

We obtain human evaluation data to validate the reliability of our factuality metric and compare it with existing ones. Human labelers assess which caption—LLaVA-v1.5-7B's or InstructBLIP's—is more factual for each image in a subset of the DOCCI test set (refer to Appendix A for further details). Using this dataset, we compare our metric with FAITHSCORE (Jing et al., 2024) and FACTSCORE (Min et al., 2023), both of which evaluate the factuality of decomposed units: FactScore uses only the reference caption, FaithScore only the image, and our metric combines both.

Table 3 shows that our metric exhibits a stronger correlation with human evaluation than existing metrics. However, this is not the sole reason to adopt our proposed metric for evaluating the factuality of detailed image captions. Metrics that rely solely on unimodal information are inherently susceptible to undesirable biases. For instance, metrics like FACTSCORE, which depend exclusively on reference cap-

Table 3: Comparison of correlations between human preferences and automated metrics in terms of factuality.

|  | FAITHSCORE | FACTSCORE | Ours |
|---|---|---|---|
| Spearman's $\rho$ | 62.5 | 67.9 | **70.2** |

tions, introduce stylistic biases tied to the specific style, tone, or phrasing of the references, unfairly favoring or penalizing captions based on these factors. In contrast, as demonstrated in Appendix B, our metric is free from such biases.

### 4.3. Improving Captioning Model Factuality

Our proposed CapMAS exhibits a loose factuality-coverage trade-off depending on the hyperparameter $\pi$. Specifically, as $\pi$ decreases, the threshold for determining factual propositions becomes stricter, leading to more propositions being identified for correction. Consequently, factuality increases while coverage decreases (an ablation study on $\pi$ is provided in Appendix C). We first investigate whether CapMAS can enhance the factuality of various MLLMs while minimizing the reduction in coverage.

Table 4 demonstrates that **CapMAS can significantly enhance the factuality of all tested MLLMs while minimizing coverage loss**. The substantial improvement in factuality, compared to the relatively minor coverage loss in the captioning models, is also reflected in the increased CLAIR scores. **Using a more advanced LLM in CapMAS does not necessarily result in greater performance gains**. When applying CapMAS to the LLaVA and InternVL models, there is minimal difference between the results obtained with LLaMA-3-8B and those with GPT-4. This suggests that the LLM's role in CapMAS is relatively straightforward. **CapMAS can improve detailed image captioning even for the state-of-the-art MLLM, GPT-4V**. It can significantly enhance factuality even when used with MLLMs far less capable than GPT-4V. However, in such cases, there is a considerable loss in coverage, as many visual elements recognized by GPT-4V are identified as hallucinations by CapMAS. With InternVL-Chat-V1.5, CapMAS maintains GPT-4V's coverage while improving factuality. We additionally provide a qualitative comparison in Figure 5 between LLaVA-NeXT-7B with and without the application of CapMAS (referencing the first two rows of Table 4).

### 4.4. Comparison with Other Methods

Various methods have been proposed to mitigate hallucinations in MLLMs, and they have primarily been validated on VQA and simple captioning benchmarks. We compare CapMAS with three recent decoding-based methods (VCD, OPERA, and SPARC (Jung et al., 2025)), two corrector-based methods (LURE and Volcano), and one training-

Table 4: Effectiveness of our proposed method across various captioning models. In the CapMAS column, the LLM represents the decomposer and corrector, while the MLLM represents the fact-checker. Avg. denotes the average of CLAIR, Factuality, and Coverage.

| Captioner | CapMAS | | Metric | | | |
|---|---|---|---|---|---|---|
| | LLM | MLLM | CLAIR | Factuality | Coverage | Avg. |
| LLaVA-NeXT-7B | - | - | 68.8 | 59.9 | **47.9** | 58.9 |
| | LLaMA-3-8B | LLaVA-NeXT-7B | 74.1 | 72.2 | 46.9 | 64.4 |
| | GPT-4 | LLaVA-NeXT-7B | **74.6** | **73.4** | 46.2 | **64.7** |
| LLaVA-NeXT-13B | - | - | 70.2 | 62.1 | **48.5** | 60.3 |
| | LLaMA-3-8B | LLaVA-NeXT-13B | **75.5** | 77.9 | 45.8 | **66.4** |
| | GPT-4 | LLaVA-NeXT-13B | 73.4 | **79.3** | 45.1 | 65.9 |
| InternVL-Chat-V1.5 | - | - | 74.9 | 65.5 | **48.2** | 62.9 |
| | LLaMA-3-8B | InternVL-Chat-V1.5 | **78.2** | **75.9** | 47.3 | **67.1** |
| | GPT-4 | InternVL-Chat-V1.5 | 77.8 | 75.7 | 47.3 | 66.9 |
| GPT-4V | - | - | 82.4 | 77.1 | **53.5** | 71.0 |
| | LLaMA-3-8B | LLaVA-NeXT-7B | 83.3 | 83.3 | 50.8 | 72.4 |
| | LLaMA-3-8B | LLaVA-NeXT-13B | 81.9 | **85.3** | 48.4 | 71.9 |
| | LLaMA-3-8B | InternVL-Chat-V1.5 | **84.6** | 82.1 | **53.5** | **73.4** |

Table 5: Performance comparison between our proposed method and other methods regarding detailed image captioning. Base refers to the default image captioning of LLaVA-v1.5-7B.

| Method | CLAIR | Factuality | Coverage | Avg. |
|---|---|---|---|---|
| Base | 62.1 | 52.8 | 34.3 | 49.7 |
| VCD (Leng et al., 2024) | 59.7 | 44.6 | 39.3 | 47.9 |
| OPERA (Huang et al., 2024) | 59.1 | 53.0 | 34.1 | 48.7 |
| LURE (Zhou et al., 2024) | 57.2 | 51.9 | 27.6 | 45.6 |
| Volcano (Lee et al., 2024) | 63.9 | 53.7 | 37.7 | 51.7 |
| LRV (Liu et al., 2023a) | 39.7 | 29.1 | 37.8 | 35.5 |
| SPARC (Jung et al., 2025) | 64.7 | 50.2 | **44.9** | 53.3 |
| CapMAS (ours) | **66.3** | **63.4** | 33.1 | **54.3** |

based method (LRV) from the perspective of detailed image captioning. All methods, except for LRV and LURE, use LLaVA-v1.5-7B, while the LRV and LURE methods employ the MiniGPT-4 model (Zhu et al., 2023) as provided by their respective authors. For reference, VisualFactChecker (VFC) (Ge et al., 2024) is also a pipeline composed of pretrained models that revise initial captions, similar to our approach. However, the inability to reproduce VFC, as its authors have not provided the necessary resources for reproduction, prevents a direct comparison with our method. Nonetheless, we can infer that our method outperforms VFC because 1) VFC specifically targets object hallucinations, and 2) it employs an object detector (Liu et al., 2023b) for hallucination detection (see Table 1).

Table 5 shows that the decoding-based methods are ineffective for detailed image captioning. Ironically, applying VCD significantly reduces the factuality of the LLaVA model while increasing coverage. SPARC does not show a notable

impact in terms of factuality, but it proves to be considerably beneficial with respect to coverage. Volcano yields only slight improvements in LLaVA's captions. However, CapMAS substantially enhances the factuality of the captioning model compared to the other methods. *These results suggest that methods proposed to enhance MLLM factuality should be evaluated not only on tasks requiring short responses, such as VQA, but also on detailed image captioning tasks.*

### 4.5. Consistency Between MLLM Captioning and VQA Evaluation results

Currently, MLLM evaluations are conducted on tasks that require only short responses, such as VQA tasks (Duan et al., 2024). However, to assess the potential of MLLMs in real-world applications, such as visual assistants, it is essential to evaluate their detailed image captioning abilities. The ranking of models used in our experiments, including LLaVA-v1.5-7B, LLaVA-NeXT-7B, LLaVA-NeXT-13B, InternVL-Chat-V1.5, and GPT-4V, is consistent across both our captioning evaluation results and widely used benchmarks like MMMU (Yue et al., 2024). However, for instance, some MLLMs may be optimized for VQA tasks that require only short responses, allowing them to rank highly on common VQA benchmarks, yet their limited image captioning abilities could restrict their practical use. To investigate this, we evaluate the detailed image captioning capabilities of various MLLMs and examine whether their rankings are consistent with their rankings on widely used VQA benchmarks. We adopt InstructBLIP-7B (Dai et al., 2023), Idefics2-8B (Laurençon et al., 2024), and MiniCPM-V-2.6 (Yao et al., 2024) as additional MLLMs for the experiment.

Table 6: Detailed image captioning and VQA performance of various MLLMs. OpenCompass (Duan et al., 2024) includes MMBench v1.1 (Liu et al., 2023c), MMStar (Chen et al., 2024a), MMMU val (Yue et al., 2024), MathVista (Lu et al., 2024), OCRBench (Liu et al., 2024d), AI2D (Kembhavi et al., 2016), HallusionBench (Guan et al., 2024), and MMVet (Yu et al., 2023). For POPE (Li et al., 2023b), we report the average F1 score across the three categories: adversarial, popular, and random. We report the sum of the perception and cognition scores for MME (Yin et al., 2023b). The best results for each metric are shown in **bold**.

| Model | Detailed Image Captioning | | | | Visual Question Answering | | | |
|---|---|---|---|---|---|---|---|---|
| | CLAIR | Factuality | Coverage | Avg. | OpenCompass | MME | POPE | Avg. |
| InstructBLIP-7B | 57.2 | 44.4 | 30.3 | 43.9 | 31.1 | 1391.4 | 86.1 | 38.4 |
| LLaVA-v1.5-7B | 61.1 | 56.3 | 30.5 | 49.3 | 36.9 | 1808.4 | 86.1 | 44.6 |
| LLaVA-NeXT-7B | 63.8 | 58.5 | 42.2 | 54.8 | 44.7 | 1769.1 | 87.5 | 50.8 |
| LLaVA-NeXT-13B | 64.5 | 62.8 | 43.0 | 56.8 | 47.6 | 1745.6 | **87.8** | 53.1 |
| Idefics2-8B | 58.1 | **85.2** | 13.4 | 52.2 | 53.0 | 1847.6 | 86.2 | 57.6 |
| InternVL-Chat-V1.5 | 72.4 | 67.6 | 46.0 | 62.0 | 61.7 | 2189.6 | 87.5 | 65.9 |
| MiniCPM-V-2.6 | 73.1 | 68.9 | 43.6 | 61.9 | **65.2** | **2268.7** | 83.2 | **68.6** |
| GPT-4V | **82.4** | 78.6 | **52.6** | **71.2** | 63.5 | 2070.2 | 81.8 | 66.4 |

Table 6 presents the evaluation results of MLLMs' responses to the prompt "Describe the given image in a very detailed manner" as well as the performance of these models on various VQA tasks. From these results, we observe that the performance of an MLLM on widely used benchmarks does not necessarily reflect its capabilities in detailed image captioning. Specifically, Idefics2-8B ranks mid-tier among the tested models in VQA tasks but falls into the lowest-performing group in terms of detailed image captioning. Its high factuality but low coverage indicates that Idefics2-8B has been trained to provide short and concise answers; this conclusion remains unchanged even when using Idefics2-8B-Chatty (Laurençon et al., 2024). Despite being a relatively small model, MiniCPM-V-2.6 attracted attention by outperforming GPT-4V on benchmarks. However, our results show that the model significantly underperforms GPT-4V in detailed image captioning. Additionally, we find that the factuality of the captions cannot be reliably predicted from the accuracy of MLLMs on POPE (Li et al., 2023b), which was proposed to evaluate object hallucinations.

*Based on these experimental results, we raise concerns about the current MLLM evaluations that are centered around VQA tasks. We encourage the community to also evaluate MLLMs from the perspective of detailed image captioning in order to showcase their full potential.*

We provide additional results on fluency, cost, and test-time scaling in Appendix F.

## 5. Conclusion

Detailed image captioning tasks are closely linked to critical applications, such as visual assistance for the impaired. Our research aims to assess and enhance the potential of MLLMs in these real-world contexts. We propose CapMAS, a method that improves detailed image captions through the collaboration of a pre-trained MLLM and LLM. In addition, we introduce a framework and benchmark dataset for evaluating the factuality and coverage of captioning models. Our experiments validate the proposed evaluation framework and demonstrate that CapMAS significantly improves the factuality of captioning models. We additionally present the following two key observations:

• Methods designed to improve MLLM factuality, which have been validated primarily on VQA or short captioning tasks, may be ineffective for detailed image captioning and can even reduce the factuality of the backbone model.
• High performance on commonly used VQA-centric benchmarks does not necessarily indicate that the model will excel in hyper-detailed image captioning.

These observations raise concerns about the current VQA-centric trend in MLLM evaluation. We encourage the community to evaluate MLLMs and related algorithms not only on VQA tasks but also on detailed image captioning tasks to gain a more comprehensive understanding of their potential.

## Impact Statement

This research contributes to developing more accurate and reliable image captioning systems, which are crucial for accessibility technologies. The proposed multiagent approach mitigates the risks of misinformation and hallucinations in AI-generated content, enhancing the safety and trustworthiness of AI systems. However, as image captioning models become more detailed, ethical concerns, particularly privacy-related, may emerge. Future research should address these challenges to ensure the responsible deployment of this technology.

## Acknowledgements

This work was supported by Institute of Information & communications Technology Planning & Evaluation (IITP) grant funded by the Korea government (MSIT) [No.RS-2021-II211343, RS-2022-II220959, Artificial Intelligence Graduate School Program (Seoul National University), No.RS-2025-02263754, Human-Centric Embodied AI Agents with Autonomous Decision-Making], the National Research Foundation of Korea (NRF) grant funded by the Korea government (MSIT) (No. 2022R1A3B1077720), a grant from Yang Young Foundation, and the BK21 FOUR program of the Education and the Research Program for Future ICT Pioneers, Seoul National University in 2025.

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

## A. Human Evaluation Dataset Construction

The results in Table 3 are obtained through the following process:

1. Captions are generated for 100 DOCCI test images using LLaVA-v1.5-7B and InstructBLIP.
2. Human labelers evaluate the captions from LLaVA-v1.5-7B and InstructBLIP for each image in terms of factuality.
3. Caption pairs with similar factuality quality are excluded.
4. For the remaining pairs, the correlation between human decisions and those made by each automated metric is measured.

## B. Undesirable Bias in FACTSCORE

Metrics that rely solely on unimodal information are inherently susceptible to undesirable biases. For instance, metrics like FACTSCORE, which depend exclusively on reference captions, introduce stylistic bias tied to the specific style, tone, or phrasing of the references, unfairly favoring or penalizing captions based on these factors. To demonstrate this, we compare FACTSCORE with our factuality metric using human-labeled captions that are hallucination-free but stylistically different from DOCCI captions (HUMAN) (Garg et al., 2024). Table 7 shows that, due to its stylistic bias, FACTSCORE assigns lower scores to these human-labeled captions, even though they are clearly superior to LLaVA-v1.5-7B and InstructBLIP captions in terms of factuality. In contrast, our factuality metric remains robust against such bias.

Table 7: Comparison of correlations between human preferences and automated metrics in terms of factuality.

| Task | Spearman's $\rho$ | |
|---|---|---|
| | FACTSCORE | Ours |
| LLaVA-v1.5-7B vs. InstructBLIP | 67.9 | 70.2 |
| HUMAN vs. LLaVA-v1.5-7B vs. InstructBLIP | 18.3 | 61.4 |

## C. Ablation Study

Table 8: Effectiveness of our proposed method across various captioning models as a function of $\pi$. In the CapMAS column, the LLM represents the decomposer and corrector, while the MLLM represents the fact-checker.

| Captioner | CapMAS | | | Metric | | |
|---|---|---|---|---|---|---|
| | LLM | MLLM | $\pi$ | CLAIR | Factuality | Coverage |
| LLaVA-NeXT-7B | - | - | - | 68.8 | 59.9 | 47.9 |
| | LLaMA-3-8B | LLaVA-NeXT-7B | 1.0 | 74.1 | 72.2 | 46.9 |
| | LLaMA-3-8B | LLaVA-NeXT-7B | 0.5 | 73.6 | 76.9 | 43.7 |
| | LLaMA-3-8B | LLaVA-NeXT-7B | 0.3 | 72.2 | 76.8 | 40.0 |
| LLaVA-NeXT-13B | - | - | - | 70.2 | 62.1 | 48.5 |
| | LLaMA-3-8B | LLaVA-NeXT-13B | 1.0 | 75.5 | 77.9 | 45.8 |
| | LLaMA-3-8B | LLaVA-NeXT-13B | 0.5 | 74.8 | 79.9 | 42.1 |
| | LLaMA-3-8B | LLaVA-NeXT-13B | 0.3 | 72.6 | 80.5 | 39.6 |
| InternVL-Chat-V1.5 | - | - | - | 74.9 | 65.5 | 48.2 |
| | LLaMA-3-8B | InternVL-Chat-V1.5 | 1.0 | 78.2 | 75.9 | 47.3 |
| | LLaMA-3-8B | InternVL-Chat-V1.5 | 0.5 | 79.0 | 78.8 | 46.0 |
| | LLaMA-3-8B | InternVL-Chat-V1.5 | 0.3 | 77.7 | 81.7 | 42.5 |

Our proposed method features a single hyperparameter, $\pi$, which serves as the threshold for classifying atomic propositions as hallucinations or non-hallucinations. Table 8 presents the effects of CapMAS across various models as a function of $\pi$. The results reveal a loose trade-off between factuality and coverage depending on $\pi$. Specifically, in all tested settings, as $\pi$ increases, factuality tends to decrease while coverage increases.

## D. The complete version of Table 2

Table 9: Meta-evaluation results across various caption evaluation methods. DOCCI and its synthetic hallucinatory captions are used for the meta-evaluation. The highest-rated caption for each method is highlighted in **bold**.

| Caption | Evaluation Metric | | | | | | | | |
|---|---|---|---|---|---|---|---|---|---|
| | BLEU | ROUGE | METEOR | CIDEr | CLIP-S | RefCLIP-S | CLAIR | ALOHa | Ours |
| Clean | 4.2 | 22.0 | 13.7 | 6.4 | 81.3 | 75.5 | **86.9** | 36.2 | **62.8** |
| Object | **4.9** | **22.3** | **14.5** | 4.8 | 81.0 | 75.3 | 85.2 | 31.5 | 52.3 |
| Attribution | 4.1 | 21.8 | 13.6 | 6.2 | 80.9 | 75.2 | 80.0 | 34.3 | 60.9 |
| Relation | 4.1 | 21.8 | 13.7 | **6.7** | **81.4** | **75.6** | 83.5 | **36.9** | 51.9 |

## E. Case

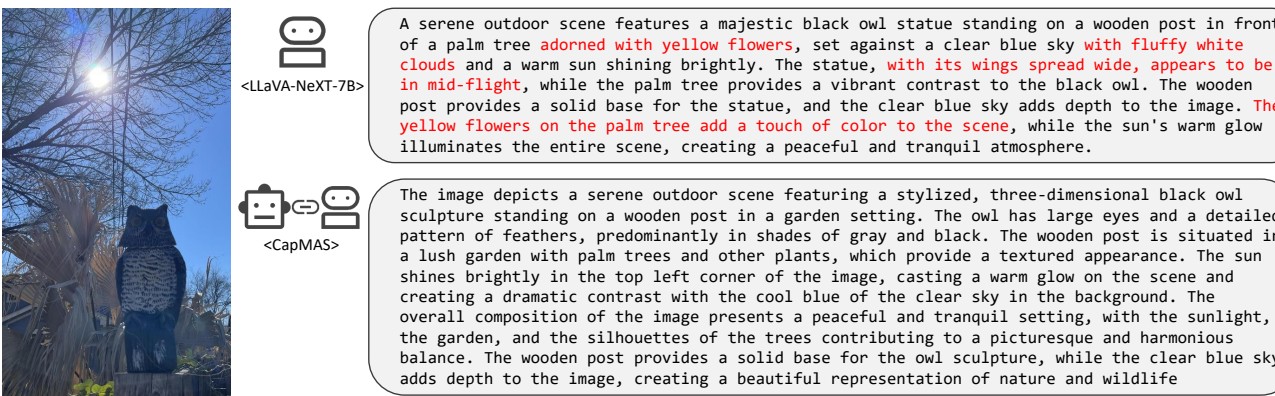

Figure 5: An example of a caption generated by CapMAS, with LLaVA-NeXT-7B as both the captioning and fact-checking model and LLaMA-3-8B as both the decomposer and corrector LLM.

## F. Additional Experiments

### F.1. Fluency Analysis

We investigate the effect of our framework on the fluency of the generated captions. We employ an LLM-based evaluation to measure the basic quality of language in the captions generated by CapMAS. Specifically, we utilize GPT-4o with the following prompt:

> You are a language expert evaluating the fluency of image captions.
>
> Fluency refers to how grammatically correct, natural, and well-formed the text sounds to a native English speaker.
> A fluent caption should be grammatically correct, free of awkward phrasing, and read smoothly.
>
> Evaluate the fluency of the following caption and return your output **strictly in JSON format** with:
> - "reason": a key reason for your scoring
> - "score": a number between 0 (completely disfluent) and 100 (perfect fluency)
>
> Caption: "{caption}"

In addition to evaluating the captions generated by CapMAS, we also assess human-written, detailed captions for the same set of images.

The results in Table 10 demonstrate that the captions generated by CapMAS achieve even higher fluency scores than the human-generated captions. This can be attributed to the final stage of CapMAS, in which the corrector LLM helps preserve or even improve the fluency of the captions.

Table 10: Fluency evaluation results. The results of CapMAS (captioner) demonstrate that the proposed framework can generate captions with higher fluency than the human-written reference captions (Human).

| Captions generated by | Fluency ↑ |
|---|---|
| Human | 89.0 |
| CapMAS (LLaVA-v1.5-7B) | 93.4 |
| CapMAS (LLaVA-NeXT-7B) | 93.4 |
| CapMAS (LLaVA-NeXT-13B) | 93.6 |
| CapMAS (InternVL-Chat-V1.5) | 94.1 |

Table 11: Effect of Self-Refine (SR) on image captioning. The results show that the quality of captions degrades as SR iterations increase.

| Method | CLAIR | Factuality | Coverage |
|---|---|---|---|
| Base | 82.4 | 77.1 | 53.5 |
| +SR (x1) | 79.9 | 72.3 | 50.4 |
| +SR (x2) | 79.1 | 70.6 | 50.1 |
| +SR (x3) | 78.5 | 69.8 | 49.7 |

### F.2. Effect of Self-Refine on Image Captioning

CapMAS can be interpreted as an inference-time scaling strategy. However, allocating more compute at inference time does not necessarily lead to better results. To demonstrate this, we test the effectiveness of inference-time scaling through Self-Refine (SR) (Madaan et al., 2023). Since validating and revising one's output requires advanced reasoning capabilities, we conducted the experiments using GPT-4V. For SR, we used the following prompt:

> You are given an image and its corresponding caption. Your task is to:
>
> 1. Analyze the image and compare it with the caption.
> 2. Identify and correct any factual or descriptive errors in the caption based on the image.
> 3. Refine the caption for clarity, correctness, and completeness — even if the original caption is mostly accurate.
>
> Show your reasoning and then provide a final refined caption.

Table 11 shows that having the model revise its own captions iteratively does not lead to better results.

### F.3. Efficiency of CapMAS

CapMAS is an approach that achieves better results by incurring additional cost at inference time. Although it involves a multi-model pipeline, CapMAS can offer a better cost-performance trade-off than Self-Refine (SR) methods for the following reasons:

- Most of CapMAS's cost lies in the final step, where the corrector LLM generates a refined caption based on the initial caption and the True/False classification results of the propositions. The decomposition step involves shorter sequences, and the MLLM used for proposition classification can process them in parallel, generating only a single token (True or False) per proposition. SR processes long sequences that include the original caption, detailed feedback, and the refined caption. Considering the length and complexity of the feedback (Madaan et al., 2023), CapMAS and SR can be seen as comparable in cost.
- CapMAS's performance does not heavily depend on the capability of the LLM used. This suggests that the LLM-related cost in the pipeline could potentially be further reduced.

## G. Prompt Templates

```
prompt_1 = "Describe the given image in a very detailed manner."
prompt_2 = "Provide a detailed description of the specified image."
prompt_3 = "Elaborate on the details of the image provided."
prompt_4 = "Offer an in-depth description of the given image."
prompt_5 = "Thoroughly describe the features of the specified image."
```

Figure 6: The five prompt inputs used to generate captions in our experiments.

**system:**
I want to verify if the given CAPTION is accurate. To assist with this verification, decompose the given CAPTION into atomic propositions. All parts of the caption must be broken down into propositions. The outputs should follow the following format:'1. proposition one\n2. proposition two\n3. proposition three'. For example, break down 'He is tall, thin, and pale' into '1. He is tall.\n2. He is thin.\n3. He is pale.'

**user:**
CAPTION: {caption}

Figure 7: The prompt input for LLaMA-3-8B serving as the decomposer.

**system:**
I want to create a caption that includes only facts. Please help me correct the given caption. The given caption contain things that are not true. Based on the given FACTS and NON-FACTS remove the non-factual elements from the caption. Place the revised caption between '###'.

**user:**
Caption: {caption}\nFACTS:\n{Non-hallucinations among the atomic propositions}\nNON-FACTS:\n{n{Hallucinations among the atomic propositions}

Figure 8: The prompt input for LLaMA-3-8B serving as the corrector.

**system:**
This is a hard problem. Carefully summarize in ONE detailed caption based on the following 5 captions by different people describing the same image. Be sure to describe everything, and avoid hallucination. Provide the detailed caption in the format '### {Detailed caption} ###'.

**user:**
Caption 1: {caption 1st}\n Caption 2: {caption 2nd}\n Caption 3: {caption 3rd}\n Caption 4: {caption 4th}\n Caption 5: {caption 5th}\n

Figure 9: The prompt input for LLaMA-3-8B serving as the summerizer. We use the prompt employed in the work of (Ge et al., 2024).

```
if hallucination == "Object"
    prompt_sys = "I want to inject incorrect information into the caption of the given photo.
Your role is to modify about THREE words from the latter part of the given caption that
describe the attributes of the objects so that they do not match the photo."
elif hallucination == "Attribution"
    prompt_sys = "I want to inject incorrect information into the caption of the given photo.
Your role is to imagine an object that isn't actually in the image but could plausibly be
there, and add a very brief part about it to the caption so that they do not match the photo."
elif hallucination == "Relation":
    prompt_sys = "I want to inject incorrect information into the caption of the given photo.
Your role is to change the spatial relationships between the objects so that they do not match
the photo. For example, change 'A person is standing to the right of the car' to 'A person is
standing to the left of the car.' Do not change anything other than the spatial relationships
between the objects."
```

```
system:
{prompt_sys}
```

```
user (presented with the image):
Caption: {caption}
```

Figure 10: The prompt input for GPT-4o used to create the meta-evaluation dataset of Table 2.

```
system:
"I want to use an object detector to check the correctness of an image caption obtained by an
image caption model. Can you help to parse the given CAPTION and list all objects that could
be detected with an object detection model in the image? Please only list the object name and
ignore the description. Please use the name in the CAPTION as it is. Please concatenate them
together with \";\" as separation."
```

```
user:
CAPTION: {caption}
```

Figure 11: The prompt input for GPT-4 used to create the dataset of Figure 1. We use the prompt employed in the work of (Ge et al., 2024).

