# OpenReview forum: "Toward Robust Hyper-Detailed Image Captioning: A Multiagent Approach and Dual Evaluation Metrics for Factuality and Coverage"
_ICML.cc/2025/Conference — ICML 2025 poster_

### Official Review · Reviewer_fLiQ · 2025-03-12

**Overall Recommendation:** 3

**Summary:**

This paper studies how to evaluate and tackle the hallucination phenomenon of MLLM. It first conduct a motivation experiments and conclude that existing hallucination detection methods struggles with long captions. Then it proposes a new multi agent approach which involves a LLM to decompose the original long detailed caption into atomic propositions, and another MLLM for fact checking. Additionally, a new evaluation metric and benchmark for factuality and coverage are proposed. Experiments on IIW-400 and DOCCI datasets with various common LLM and MLLM demonstrate the effectiveness of the proposed approach.

**Claims And Evidence:**

Yes, the motivation justified by experiments in section 3.1 is straightforward and makes intuitive sense to me. Long-context understanding has always been a problem for LLM/MLLM, and it naturally comes to one's mind that hallucination detectors would lean more on closer text and less on far-way image when the generated caption is extremely long.

**Essential References Not Discussed:**

N/A

**Experimental Designs Or Analyses:**

Experiments in Table 4 does show non-trivial improvement when the proposed CapMAS is applied. However, with a stronger captioner like GPT-4V, the gain becomes smaller. Thus, one wonders if a truly powerful MLLM with superb long-context understanding capability is employed, how much gain would CapMAS bring?

Secondly, how does the corrector LLM takes as input the $\pi$-thresholding results and correct the original caption? I wonder have the authors check if the hallucination scores would continue to improve if CapMAS is applied iteratively?

**Methods And Evaluation Criteria:**

Yes. Breaking the detailed caption into atomic propositions naturally tackles the long-context understanding issue.

I wonder, do the authors have some experimental evidences to prove that after CapMAS correction, the new caption is on par with or better than the original caption besides hallucination degree? A high-quality image caption should not hallucinates, and also truly and faithfully describe the whole image content in fluent human-style language.

**Other Comments Or Suggestions:**

N/A

**Other Strengths And Weaknesses:**

This CapMAS approach requires two extra LLM and one MLLM. I wonder, what are the memory and computational cost of CapMAS when compared with naive baseline and other hallucination tackling methods? Could this strategy be interpreted as some kind of inference-time scaling? If so, how well does CapMAS perform when compared to simple inference-time scaling strategy of MLLM image captioning?

**Questions For Authors:**

N/A

**Relation To Broader Scientific Literature:**

Tackling LLM/MLLM hallucination is an important aspect of current foundational model researches.

**Theoretical Claims:**

N/A

---

> ### Author Rebuttal · Authors · 2025-04-01
>
> **We are pleased to share that, in support of open-source research, we have decided to release our carefully curated VQA dataset and evaluation codes. This dataset includes 1k images, each paired with approximately 36 question-answer sets.**
>
> We sincerely thank you for your thorough review. We have made our best efforts to address your remaining concerns as follows:
>
> >**Q1**. Do the authors have some experimental evidence to prove that after CapMAS correction, the new caption is on par with or better than the original caption besides the hallucination degree?
>
> **A1**. We sincerely thank the reviewer for suggesting this experiment, which allowed us to highlight a new strength of CapMAS in terms of fluency. Due to space limitations, we regret that we cannot provide a detailed response here, and kindly ask you to refer to our response **A2** to reviewer **quCw** for further details.
>
> >**Q2**. Thus, one wonders if a truly powerful MLLM with superb long-context understanding capability is employed, how much gain would CapMAS bring?
>
> **A2**. CapMAS is designed to improve the captioning performance of MLLMs, which currently have limited long-context capabilities and are prone to hallucinations. As such, its effect may be less pronounced for future MLLMs with powerful capabilities. However, it is important to highlight that even for GPT-4V, one of the current SOTA MLLMs, we were able to improve factuality by over 5% points using CapMAS—without sacrificing coverage—by leveraging relatively weaker open-source models. Given that today’s best MLLMs still fall short of a truly reliable MLLM, our work carries meaningful implications, particularly for visually impaired users.
>
> >**Q3**. How does the corrector LLM take as input the $\pi$-thresholding results and correct the original caption?
>
> **A3**. It can be easily understood by referring to the corrector LLM's prompt template:
> ```
> system:
> I want to create a caption that includes only facts. Please help me correct the given caption.
> The given caption contains things that are not true. Based on the given FACTS and NON-FACTS,
> remove the non-factual elements from the caption.
>
> user:
> Caption: {caption}
> FACTS:
> {propositions classified as non-hallucinatory}
> NON-FACTS:
> {propositions classified as hallucinatory}
> ```
> >**Q4**. I wonder have the authors check if the hallucination scores would continue to improve if CapMAS is applied iteratively?
>
> **A4**. CapMAS is designed to remove detected hallucinations rather than correct them. This design choice is based on our observation that attempts at correction often lead to the introduction of new hallucinations. As a result, applying CapMAS iteratively to a single caption has an effect equivalent to applying it once with a lower threshold $\pi$.
>
> Instead, we demonstrate that applying CapMAS individually to multiple captions for a single image and subsequently summarizing the results can lead to an overall improvement in the caption quality for that image.
>
> Table B: Results of LLaVA-Next on a subset of IIW400
>
> |Number of CapMAS Applications|CLAIR|Factuality|Coverage|
> |-|-|-|-|
> |1|59.8|83.0|24.3|
> |2|63.3|76.8|32.6|
> |3|67.4|80.2|37.7|
> |4|68.2|79.9|38.9|
> |5|70.6|80.1|41.4|
>
> Table B demonstrates that the overall quality of the resulting captions improves as the number of CapMAS applications increases.
>
> >**Q5**. I checked Appendix D and found that a larger threshold generally lead to better performances. I wonder, will it eventually plateau?
>
> **A5**. In CapMAS, as the value of $\pi$ increases, the criterion for judging a proposition as true becomes more relaxed. This results in improved coverage but also causes more propositions to be classified as true, thereby introducing more hallucinations and lowering factuality. Consequently, in Appendix D, further increases in $\pi$ lead to a significant drop in the factuality score, ultimately degrading the overall quality of the captions. We summarize the results of additional experiments in the following table:
>
> Table C: Ablation results on $\pi$ using LLaVA-NeXT-7B
>
> |$\pi$|CLAIR|Factuality|Coverage|
> |-|-|-|-|
> |3.0|71.1|63.0|47.8|
> |2.0|70.6|65.0|47.1|
> |1.0|74.1|72.2|46.9|
> |0.5|73.6|76.9|43.7|
>
> >**Q6**. Could this strategy be interpreted as some kind of inference-time scaling?
>
> **A6**. Yes, CapMAS can indeed be considered an inference-time scaling strategy, and the experimental results, including those in Table B, support this interpretation. Additionally, we compared CapMAS with Self-Refine, another inference-time scaling method. CapMAS achieves significantly better performance while requiring a comparable level of cost. Due to space limitations, we regret that we cannot provide further details here and kindly ask you to refer to our responses **A3 and A4** to reviewer **bFJq** for more information.
>
> **We have additional experimental results that we were unable to include here. To help us share them, we kindly ask you to click “Rebuttal Comment” to allow us to leave a supplementary comment.**

---

> > ### Comment · Reviewer_fLiQ · 2025-04-04
> >
> > Please share your additional results in a supplementary comment. Thanks!

---

> > > ### Author Response · Authors · 2025-04-08
> > >
> > > Thank you for your comment!
> > >
> > > >**Q7**. How well does CapMAS perform when compared to simple inference-time scaling strategy of MLLM image captioning?
> > >
> > > **A7**. Motivated by the reviewer’s comment, we tested the effectiveness of inference-time scaling via Self-Refine (SR) [1]. Since validating and revising one's output requires advanced reasoning capabilities, we conducted the experiments using GPT-4V. For SR, we used the following prompt:
> > > ```
> > > You are given an image and its corresponding caption.
> > > Your task is to:
> > > 1. Analyze the image and compare it with the caption.
> > > 2. Identify and correct any factual or descriptive errors in the caption based on the image.
> > > 3. Refine the caption for clarity, correctness, and completeness — even if the original caption is mostly accurate.
> > >
> > > Show your reasoning and then provide a final refined caption.
> > > ```
> > > <Table D>
> > >
> > > |Method|CLAIR|Factuality|Coverage|
> > > |-|-|-|-|
> > > |Base|82.4|77.1|53.5|
> > > |+SR (x1)|79.9|72.3|50.4|
> > > |+SR (x2)|79.1|70.6|50.1|
> > > |+SR (x3)|78.5|69.8|49.7|
> > >
> > > Table D shows that having the model revise its own captions iteratively does not lead to better results.
> > >
> > > CapMAS is an approach that achieves better results by incurring additional cost at inference time. Although it involves a multi-model pipeline, CapMAS can offer a better cost-performance trade-off than Self-Refine methods for the following reasons:
> > > 1. Most of CapMAS’s cost lies in the final step, where the corrector LLM generates a refined caption based on the initial caption and the True/False classification results of the propositions. The decomposition step involves shorter sequences, and the MLLM used for proposition classification can process them in parallel, generating only a single token (True or False) per proposition. SR processes long sequences that include the original caption, detailed feedback, and the refined caption. Considering the length and complexity of the feedback [1], CapMAS and SR can be seen as comparable in cost.
> > > 2. As shown in Table 4 of our manuscript, CapMAS’s performance does not heavily depend on the capability of the LLM used. This suggests that the LLM-related cost in the pipeline could potentially be further reduced.
> > >
> > > [1] SELF-REFINE: Iterative Refinement with Self-Feedback
> > >
> > > **Evaluation of CapMAS on an Additional Dataset**. To demonstrate the generalizability of CapMAS, we additionally tested its effectiveness on a subset of DOCCI (400 samples).
> > >
> > > <Table E>
> > >
> > > |Method|CLAIR|Factuality|Coverage|
> > > |-|-|-|-|
> > > |LLaVA-NeXT-7b|68.0|56.2|54.0|
> > > | +CapMAS|72.8|68.1|53.5|
> > > |LLaVA-NeXT-13b|70.1|58.2|55.6|
> > > | +CapMAS|74.5|74.6|52.3|
> > >
> > > Table E demonstrates that CapMAS is also effective on DOCCI.
> > >
> > > We sincerely appreciate your thoughtful feedback and have done our best to address your concerns. If there are any remaining or additional questions, please don’t hesitate to let us know. If you find our response satisfactory, we would be grateful if you could consider reflecting that in your score.
> > >
> > > Thank you again for your time and consideration.

---

### Official Review · Reviewer_bFJq · 2025-03-12

**Overall Recommendation:** 4

**Summary:**

This paper looks at preventing hallucination in long-form image captions, proposing a system "CapMAS" which decomposes generated captions into atomic statements, which are then generated/corrected using a VLM. The paper also introduces two metrics for image caption evaluation based on a similar pipeline: Factuality (which represents the portion of "true" atomic statements), and Coverage (which uses a dataset of human annotated examples to evaluate if a caption can be used to answer all questions in an images). The paper shows that the resulting factuality metric correlates better with human judgements than both FAITHSCORE and FACTSCOR. The paper then shows that CapMAS outperforms base models alone when applied across several base captioning models (on CLAIR, Factuality, and Coverage scores), and significantly improves performance compared to hallucination reduction methods.

## update after rebuttal

Thanks to the authors for providing some clarifications to my comments - I particularly appreciate the additional discussion comparing to SR, and would love to see this included in the paper (A3). In general, while the paper is similar to some released methods, and I agree with reviewer QHjW that it is unlikely to have broad impact due to it's limited scope, it's a solid paper.

**Claims And Evidence:**

The paper makes several well-supported claims:
- CapMAS improves on factuality/clair/coverage averages in the DOCCI/ IIW-400 dataset
- VQA benchmarks do not correlate with captioning performance
- Existing hallucination detection methods fail on long captions (Though it's worth noting that CLAIR appears to be as effective in the meta-evaluation in Table 2, and ALOHa performs well in Table 2 for object-hallucination, which it is designed for).

The claim on L346 (Sec 4.3) that CapMAS exhibits a factuality-coverage tradeoff is somewhat over-stated. While the factuality and coverage scores do indeed have opposite trends in Table D, it doesn't seem to me that these are inherently conflicting values (and indeed, this is just an artifact of the fact that the LLM cannot achieve well-grounded performance).

**Essential References Not Discussed:**

The references are generally sufficient, however the paper might consider discussing a comparison to Woodpecker [1], which is a method similar to LURE for caption hallucination reduction which detects captions using an external model, and corrects the caption using information from this process. The system prompt in Figure 9 is very closely related to the system prompt in [2], and should probably be cited. The paper could consider citing related work in NLP on self-rationalization models which look at breaking sentences down into atomic claims [3].

[1] Yin, Shukang, et al. "Woodpecker: Hallucination correction for multimodal large language models." Science China Information Sciences 67.12 (2024): 220105.
[2] Chan, David, et al. "IC3: Image Captioning by Committee Consensus." The 2023 Conference on Empirical Methods in Natural Language Processing.
[3] Wiegreffe, Sarah, and Ana Marasović. "Teach me to explain: A review of datasets for explainable natural language processing." arXiv preprint arXiv:2102.12060 (2021).

**Experimental Designs Or Analyses:**

I did not validate the experimental design, however no tables in the paper have any estimate of variance (for example, standard error around the mean), making it challenging to determine which experiments have significant effects. Also notable is that there is little ablation of the CapMAS components, for example, is it necessary to break the caption down into atomic points before re-captioning, or is just asking the external VLM to re-caption sufficient?

**Methods And Evaluation Criteria:**

The methods/evaluation criteria are fairly well-designed, though I would like to see some experiments on a wider set of images (particularly compared to IIW-400 which is quite small at only 400 images). Experimental comparisons on COCO (though less centered on detail captions), would be quite helpful in placing this against other captioning methods.

**Other Comments Or Suggestions:**

- It would be quite helpful to have some additional qualitative analysis, or error analysis, to look at potential directions for future research, or to evaluate weaknesses in the approach (for both CapMAS and for the evaluation metrics).

**Other Strengths And Weaknesses:**

Strengths:
- The paper is quite strong in terms of performance on the discussed metrics, and clearly leads to overall improvements on a limited data subset.
- The paper requires no additional training (is fully zero-shot)
- Factuality seems to be quite good compared to both baseline measures

Weaknesses:
- As discussed above, the test datasets used here are fairly small and limited in scope, and it would be good to see evaluations on something a bit broader.
- There's not really any explanation or discussion of the efficiency of the model, which now requires several LLMs (and probably significantly increases the cost of generating individual captions)

**Questions For Authors:**

- I couldn't find the dataset used for evaluation in Tables 4 and 5, is this on  IIW-400 or the DOCCI dataset?

**Relation To Broader Scientific Literature:**

This paper consists of two key components, the method CapMAS and the evaluation measures/dataset. The evaluation measures are, while useful, quite similar to related work. Coverage is quite similar in structure to the POPE metric, albeit with a different goal (rather than hallucination detection, image factuality). Factuality is quite similar to the ALOHa metric, but expands somewhat from objects to atomic statements (though the motivating experiments seem to use a version that is object-detector based, which aligns quite well with ALOHa's contribution). CapMAS itself is quite similar to methods employed by both LURE and Woodpecker (See below), and uses rewriting models to directly rewrite output captions (though here, the focus is on coverage/correctness compared to hallucination).

While the approach is quite similar to existing methods, this appears to be the first approach which puts these all together, and the simplicity of the combination is quite compelling. The performance seems strong, and demonstrates that many of the ideas introduced in prior work can be combined to strong effect.

**Theoretical Claims:**

This paper makes no theoretical claims.

---

> ### Author Rebuttal · Authors · 2025-04-01
>
> **We are pleased to share that, in support of open-source research, we have decided to release our carefully curated VQA dataset and evaluation codes. This dataset includes 1k images, each paired with approximately 36 question-answer sets.**
>
> We sincerely thank you for your thorough review. We have made our best efforts to address your remaining concerns as follows:
> >**Q1**. The claim on L346 (Sec 4.3) that CapMAS exhibits a factuality-coverage tradeoff is somewhat over-stated.
>
> **A1**. CapMAS is designed to remove detected hallucinations rather than correct them. This design choice is based on our observation that attempts at correction often lead to the introduction of new hallucinations. As CapMAS may remove not only hallucinations but also factual content, it inherently involves a trade-off between factuality and coverage, as discussed in Section 4.3.
>
> Nevertheless, through inference-time scaling, CapMAS can improve the coverage of captions while maintaining a certain level of factuality. Due to space limitations, we regret that we cannot provide further details here and kindly ask you to refer to our response **A4** to reviewer **fLiQ** for more information.
> >**Q2**. I would like to see some experiments on a wider set of images.
>
> **A2**. Using benchmarks like COCO for evaluating detailed image captioning can lead to misleading conclusions:
> 1. Short captions in COCO bias evaluation metrics to favor simpler captions over accurate, detailed ones.
> 2. Reference-free metrics, intended to remove this bias, often introduce another bias stemming from the evaluation model itself, which can significantly affect the scores [1].
>
> We believe reliable evaluation requires detailed human supervision. To address your concern, we conducted additional experiments on DOCCI. Due to space limitations, we kindly ask you to refer to our response **A2** to reviewer **QHjW** for details, which confirm CapMAS is effective on DOCCI as well.
>
> [1] LLM Evaluators Recognize and Favor Their Own Generations
> >**Q3**. Is it necessary to break the caption down into atomic points before re-captioning, or is just asking the external VLM to re-caption sufficient?
>
> **A3**. CapMAS can be understood as an inference-time scaling strategy. Motivated by the reviewer’s comment, we tested the effectiveness of inference-time scaling via Self-Refine (SR) [2]. Since validating and revising one's output requires advanced reasoning capabilities, we conducted the experiments using GPT-4V. For SR, we used the following prompt:
> ```
> You are given an image and its corresponding caption.
> Your task is to:
> 1. Analyze the image and compare it with the caption.
> 2. Identify and correct any factual or descriptive errors in the caption based on the image.
> 3. Refine the caption for clarity, correctness, and completeness — even if the original caption is mostly accurate.
>
> Show your reasoning and then provide a final refined caption.
> ```
> <Table A>
>
> |Method|CLAIR|Factuality|Coverage|
> |-|-|-|-|
> |Base|82.4|77.1|53.5|
> |+SR (x1)|79.9|72.3|50.4|
> |+SR (x2)|79.1|70.6|50.1|
> |+SR (x3)|78.5|69.8|49.7|
>
> Table A shows that having the model revise its own captions iteratively does not lead to better results.
>
> [2] SELF-REFINE: Iterative Refinement with Self-Feedback
> >**Q4**. There's not really any explanation or discussion of the efficiency of the model
>
> **A4**. CapMAS is an approach that achieves better results by incurring additional cost at inference time. Although it involves a multi-model pipeline, CapMAS can offer a better cost-performance trade-off than Self-Refine (SR) methods for the following reasons:
> 1. Most of CapMAS’s cost lies in the final step, where the corrector LLM generates a refined caption based on the initial caption and the True/False classification results of the propositions. The decomposition step involves shorter sequences, and the MLLM used for proposition classification can process them in parallel, generating only a single token (True or False) per proposition. SR processes long sequences that include the original caption, detailed feedback, and the refined caption. Considering the length and complexity of the feedback [2], CapMAS and SR can be seen as comparable in cost.
> 2. As shown in Table 4, CapMAS’s performance does not heavily depend on the capability of the LLM used. This suggests that the LLM-related cost in the pipeline could potentially be further reduced.
>
> >**Q5**. I couldn't find the dataset used for evaluation in Tables 4 and 5, is this on IIW-400 or the DOCCI dataset?
>
> **A5**. For Tables 4 and 5, we used IIW-400, based on findings [3] indicating that IIW-400 serves as a better reference caption set than DOCCI. As noted in A2, our additional experimental results confirm that CapMAS is effective on both IIW-400 and DOCCI.
>
> [3] ImageInWords: Unlocking Hyper-Detailed Image Descriptions
>
> **We have additional results and discussions that we were unable to include here. We kindly ask you to click “Rebuttal Comment” to allow us to add a comment.**

---

### Official Review · Reviewer_QHjW · 2025-03-14

**Overall Recommendation:** 4

**Summary:**

This paper focuses on generating long, detailed captions for images. A key idea in the paper is to decompose long captions into atomic claims using an LLM, and then verify every claim individually in the context of the image using a VLM. The paper motivates this by showing that this approach outperforms alternative ways of identifying hallucinations, like token confidence or consistency-based approaches. Based on this observation, the paper proposes a pipeline called CapMAS, which decomposes a long caption into atomic claims, verifies the correctness of each claim, and removes incorrect claims from the caption using an LLM. The paper also proposes two metrics to measure factuality and coverage of claims in captions by using GPT-4o to break down captions into individual claims and verifying them. The paper compares these metrics to prior metrics like ROGUE and BLEU, and shows that it outperforms them wrt finding the correct caption for an image. Using these metrics, they evaluate CapMAS and prior approaches for the task of long-caption generation. CapMAS performs quite well on a wide range of VLMs, outperforming prior approaches. Finally, the paper highlights that current models are typically trained to generate short, concise responses and struggle with longer responses.

## Update after rebuttal
I think this is a solid paper. Thus, I increased my score to Accept. I hope the added results and discussions are reflected in the final version.

**Claims And Evidence:**

The claims are mostly supported by empirical evidence:
- The paper motivated the use of decomposing long captions into atomic claims by showing that it is better at detecting hallucinations as compared to simple baselines like token confidence or consistency-based approaches. This forms the basis of their pipeline, CapMAS.
- The paper proposed a metric for measuring factuality and showed that it is better than prior metrics for this purpose.
- They also proposed a new metric for measuring coverage, but did not compare it to any prior metrics for this purpose – *are there any relevant alternatives?*
- They evaluated their pipeline CapMAS using these metrics and demonstrated that it consistently improved factuality for a range of MLLMs (like LLaVA and GPT-4o), supporting the efficacy of their approach.
- They also demonstrated that models which are good at visual question answering might not necessarily be good at detailed image captioning as they might be biased toward shorter responses, reducing their coverage.

**Essential References Not Discussed:**

Mostly, the literature is well-covered.
You could additionally discuss and compare against [1] as a baseline.

[1] Petryk et al. Simple Token-Level Confidence Improves Caption Correctness. WACV'24

**Experimental Designs Or Analyses:**

Yes, the experiments in the paper are mostly sound. The factuality metric is compared to multiple existing metrics on 3 types of perturbed captions that contain hallucinations, and it is the most accurate at identifying captions across all of them. Further, the CapMAS pipeline is also compared against prior works on a variety of MLLMs and shows strong performance consistently.

*One major question is: which datasets do Tables 4 and 5 correspond to? This wasn’t clear, and is an important detail.*

**Methods And Evaluation Criteria:**

Yes. The paper focuses on removing hallucinations from detailed captions. They identify two key criteria for this setting: factuality and coverage. They propose metrics for these criteria and demonstrate their efficacy. They evaluate their approach on two datasets – DOCCI and IIW-400 – which contain images and corresponding detailed, factual captions. They test their method on a wide range of MLLMs to assess how general it is.

**Other Comments Or Suggestions:**

You can consider giving your proposed metrics a specific name (for example, the title of Section 4.2 reads a bit weird to me). Also, clearly highlight that this metric uses GPT-4o’s vision capabilities as well (i.e., there is no other model serving as the VLM, only GPT-4o).

**Other Strengths And Weaknesses:**

Strengths:
- The proposed pipeline performs well on the task of reducing hallucinations in detailed captions. It leads to significant improvement in performance for a range of MLLMs (small models like LLaVA to large models like GPT-4v). It also outperforms prior approaches.
- The pipeline can be easily plugged into existing MLLMs without requiring any additional training.
- The paper is mostly well-written and easy to follow.

Weaknesses:
- A few details were not clear. Which datasets do Tables 4 and 5 correspond to? This wasn’t clear, and is an important detail.
- While this approach helps improve factuality, it doesn’t seem to improve the coverage of captions, which is also important for detailed captions. (I understand that one can tune the hyperparameter pi to change coverage but coverage is still upper-bounded by the original captions generated by the MLLM).
- This is a minor point, but for the sake of completeness, you could also evaluate your pipeline on GPT-4o, as it is more widely used than GPT-4v.

**Questions For Authors:**

- How is coverage computed? I couldn’t find a clear formula for this in the paper.
- I believe this pipeline cannot improve the coverage in captions (and only improve factuality). How can we improve coverage?
- For measuring factuality, your method uses the image and the reference caption. If the reference caption is already available, is the image really required? Can’t you verify the claim against the reference caption? An ablation about this might be interesting, as removing the image can save significant amounts of compute/ API credits.
- Similarly, for coverage, the questions are generated based on the image. The coverage of these questions themselves would be bottlenecked by the vision capabilities of GPT-4o, and the questions might miss some details. Could you try incorporating the captions of images in the question-generation pipeline?

**Relation To Broader Scientific Literature:**

Prior literature in vision-language understanding has focused less on highly detailed captions. While approaches have been proposed to reduce hallucinations in shorter responses, the paper shows that these approaches don’t work well for longer captions. They propose a new method to improve the factuality of detailed captions and show that it outperforms existing approaches.

**Theoretical Claims:**

N/A

---

> ### Author Rebuttal · Authors · 2025-04-01
>
> **We are pleased to share that, in support of open-source research, we have decided to release our carefully curated VQA dataset and evaluation codes. This dataset includes 1k images, each paired with approximately 36 question-answer sets.**
>
> We sincerely thank you for your thorough review. We have made our best efforts to address your remaining concerns as follows:
> >**Q1**. They also proposed a new metric for measuring coverage, but did not compare it to any prior metrics for this purpose – are there any relevant alternatives?
>
> **A1**. There have been prior attempts to measure the coverage of image captions [1]. These methods typically involve extracting visual entities from reference captions and checking whether they appear in the caption being evaluated. However, such approaches have notable limitations:
> 1. MLLMs can describe the same visual content in highly diverse ways. As a result, it is often difficult to determine with precision whether the visual content in the reference captions is present in the evaluated caption.
> 2. These methods can only evaluate pre-defined types of visual content—typically objects, attributes, and spatial relations—since they rely on accurate extraction and comparison of such elements within captions. As a result, they cannot assess whether a caption captures conceptual aspects of an image, such as mood or semantic relationships between entities.
>
> In contrast, our coverage metric avoids these limitations. Any information can be turned into a question and given to an LLM grounded in the caption.
>
> [1] Object Hallucination in Image Captioning
> >**Q2**. Which datasets do Tables 4 and 5 correspond to?
>
> **A2**. For Tables 4 and 5, we used IIW-400, based on findings [2] indicating that IIW-400 serves as a better reference caption set than DOCCI. However, to demonstrate the generalizability of CapMAS, we additionally tested its effectiveness on a subset of DOCCI (400 samples).
>
> <Table A>
>
> |Method|CLAIR|Factuality|Coverage|
> |-|-|-|-|
> |LLaVA-NeXT-7b|68.0|56.2|54.0|
> | +CapMAS|72.8|68.1|53.5|
> |LLaVA-NeXT-13b|70.1|58.2|55.6|
> | +CapMAS|74.5|74.6|52.3|
>
> Table A demonstrates that CapMAS is also effective on DOCCI.
>
> [2] ImageInWords: Unlocking Hyper-Detailed Image Descriptions
> >**Q3**. I believe this pipeline cannot improve the coverage in captions (and only improve factuality). How can we improve coverage?
>
> **A3**. Through inference-time scaling, CapMAS can improve the coverage of captions while maintaining a certain level of factuality. Due to space limitations, we regret that we cannot provide further details here and kindly ask you to refer to our response **A4** to reviewer **fLiQ** for more information.
> >**Q4**. This is a minor point, but for the sake of completeness, you could also evaluate your pipeline on GPT-4o, as it is more widely used than GPT-4v.
>
> **A4**. We used GPT-4o to evaluate the captions, and it is known that LLMs tend to favor their own outputs [3]. Therefore, to ensure a fair comparison, we did not use GPT-4o at any stage of the proposed captioning pipeline.
>
> [3] LLM Evaluators Recognize and Favor Their Own Generations
> >**Q5**. How is coverage computed? I couldn’t find a clear formula for this in the paper.
>
> **A5**. We apologize for the confusion. Let N be the total number of VQA samples and C the number correctly answered by GPT-4o using only the captions. Then, the coverage score is C/N.
> >**Q6**. For measuring factuality, your method uses the image and the reference caption. If the reference caption is already available, is the image really required?
>
> **A6**. Metrics based solely on reference captions are prone to stylistic bias, favoring or penalizing captions based on phrasing. To illustrate this, we compare a reference-based metric with our factuality metric using human-labeled captions (HUMAN) that are hallucination-free but stylistically different from the references. As shown in Table B, the reference-based metric correlates poorly with human judgment by assigning lower scores to HUMAN while our factuality metric remains robust.
>
> Table B: Meta-evaluation with model- and human-generated captions; see Sec. 4.2 and Appx. B for details.
>
> |Method|Correlation with Human Evaluation &#8593;|
> |-|-|
> |Reference-based|18.3|
> |Ours|61.4|
> >**Q7**. The coverage of these questions themselves would be bottlenecked by the vision capabilities of GPT-4o, and the questions might miss some details.
>
> **A7**. Building a detailed VQA dataset is costly, so we relied on GPT-4o, making our dataset quality depend on its capabilities. To address this, we carefully guided human annotators during the labeling process.
>
> While we considered using reference captions, we first aimed to show that our coverage metric works reliably without them, as reliance on references limits its applicability. Our results confirm this, and we are exploring ways to further improve the method by incorporating reference captions.
>
> **We have more results—please click “Rebuttal Comment” to allow us to add a comment.**

---

> > ### Comment · Reviewer_QHjW · 2025-04-03
> >
> > Thanks for your efforts in the rebuttal period. A lot of my concerns were addressed. Here are more specific comments:
> >
> > Q1: While these arguments make sense, it would be good to include a quantitative comparison of the proposed coverage metric with prior work you cited [1].
> >
> > Q2: This makes sense. Please update the submission to explicitly state which dataset the table corresponds to, as well as the new results.
> >
> > Q3: This is an interesting result! You can include it in the main text.
> >
> > Q4: Okay, that makes sense.
> >
> > Q5: Thanks for the clarification; please add this to the submission.
> >
> > Q6: Thanks for the additional results. It’s surprising to see the reference-based approach perform so poorly. If you plan to add this to your paper, including a few qualitative examples of failure modes of the reference-based approach might also be helpful.
> >
> > Q7: Okay, that makes sense.
> >
> > Overall, the paper is above the threshold for acceptance; hence, I maintain my original score of Weak Accept.
> >
> > [1] Object Hallucination in Image Captioning

---

> > > ### Author Response · Authors · 2025-04-08
> > >
> > > Thank you for the comment! We will incorporate all of your suggestions into the next version of the paper. The comparison with the prior coverage metric requires manually identifying and organizing visual entities from the reference captions, so it will take some time. We will also include the results of this experiment in the next version.
> > >
> > > In this additional comment, we would like to highlight a new strength of CapMAS in terms of fluency.
> > > To assess the fluency of the captions generated by CapMAS, we employ an LLM-based evaluation. Specifically, we utilize GPT-4o with the following prompt:
> > > ```
> > > You are a language expert evaluating the fluency of image captions.
> > >
> > > Fluency refers to how grammatically correct, natural, and well-formed the text sounds to a native English speaker. A fluent  caption should be grammatically correct, free of awkward phrasing, and read smoothly.
> > >
> > > Evaluate the fluency of the following caption and return your output **strictly in JSON format** with:
> > > - "reason": a key reason for your scoring
> > > - "score": a number between 0 (completely disfluent) and 100 (perfect fluency)
> > >
> > > Caption: "{caption}"
> > > ```
> > > In addition to evaluating the captions generated by CapMAS, we also assess human-written detailed captions for the same set of images.
> > >
> > > **Table C**
> > >
> > > |Captions generated by|Fluency &#8593;|
> > > |-|-|
> > > |Human|89.0|
> > > |CapMAS (LLaVA-v1.5-7B)|93.4|
> > > |CapMAS (LLaVA-NeXT-7B)|93.4|
> > > |CapMAS (LLaVA-NeXT-13B)|93.6|
> > > |CapMAS (InternVL-Chat-V1.5)|94.1|
> > >
> > > The results in Table C demonstrate that the captions generated by CapMAS achieve even higher fluency scores than the human-generated captions. This can be attributed to the final stage of CapMAS, in which the corrector LLM helps preserve or even improve the fluency of the captions.

---

### Official Review · Reviewer_quCw · 2025-03-14

**Overall Recommendation:** 3

**Summary:**

This paper proposes a multiagent approach that leverages LLM and MLLM to correct given captions and designs two metrics for evaluating generated caption. A dataset is collected for one of the metrics.

**Claims And Evidence:**

Yes.

**Essential References Not Discussed:**

None.

**Experimental Designs Or Analyses:**

In Table 2, it appears that the CLAIR metric can reflect the introduced noises. It's not as problematic as the article suggests.

**Methods And Evaluation Criteria:**

- In Table 4, after the captions were corrected, the coverage score decreased. This may because: 1) the design of the metric is not suitable, 2) the method sacrifices coverage for factuality, indicating that the method has some drawbacks. Same phenomenon is observed in Table 5, please provide more explanation.
- Since Factuality and Coverage cannot reflect the basic quality of language, how about including the traditional metrics like CIDEr and METEOR?

**Other Comments Or Suggestions:**

None.

**Other Strengths And Weaknesses:**

- Strengths:
  - The proposed method enhances the factual accuracy of captions.
  - Existing approaches require the corrector model training, while the proposed method employs collaboration between an MLLM and LLM.
- Weaknesses:
  - Previous studies focus on measuring the factuality of generated text, while the coverage metric is newly proposed. However, the rationale for using this metric is questionable, as the two metrics appear to be contradictory based on the experimental results. Or if there is no problem with the metric, does this mean that the proposed method may not work as intended?

**Questions For Authors:**

See weaknesses.

**Relation To Broader Scientific Literature:**

This paper is related to helping MLLMs generating highly detailed captions.

**Theoretical Claims:**

Not Applicable.

---

> ### Author Rebuttal · Authors · 2025-03-31
>
> Thank you for your thoughtful review. We appreciate your recognition of our method’s potential to improve the factual accuracy of image captions and our novel approach leveraging MLLM–LLM collaboration without training a separate corrector. We also thank you for your suggestions, which have enriched our work.
>
> **We are pleased to share that, in support of open-source research, we have decided to release our carefully curated VQA dataset and evaluation codes. This dataset includes 1k images, each paired with approximately 36 question-answer sets. We kindly ask that you consider this a contribution to the open-source community.**
>
> In the subsequent sections, we address each of your concerns as follows:
>
> >**Q1**. In Table 4, after the captions were corrected, the coverage score decreased. This may because: 1) the design of the metric is not suitable, 2) the method sacrifices coverage for factuality, indicating that the method has some drawbacks. Same phenomenon is observed in Table 5, please provide more explanation.
>
> **A1**. We would like to clarify a potential misunderstanding regarding the decrease in the coverage score caused by the application of CapMAS. First and foremost, CapMAS was proposed to enhance the factuality of detailed image captions. Ideally, identifying and correcting hallucinations within captions would improve both factuality and coverage. However, through empirical analysis, we observed that such correction attempts often lead to the generation of new hallucinations due to the limitations of current MLLMs.
> Therefore, CapMAS is designed to remove detected hallucinations rather than correct them. As CapMAS may remove not only hallucinations but also factual content, it inherently involves a trade-off between factuality and coverage, as discussed in Section 4.3. This trade-off is controlled by its hyperparameter, $\pi$.
> Consequently, the decrease in coverage scores observed in Tables 4 and 5 arises not from a flaw in the proposed metrics, but rather from the design of CapMAS, which reflects the current limitations of MLLMs.
>
> >**Q2**. Since Factuality and Coverage cannot reflect the basic quality of language, how about including the traditional metrics like CIDEr and METEOR?
>
> **A2**. We employ an LLM-based evaluation to measure the basic quality of language in the captions generated by CapMAS. This approach is motivated by recent studies [1,2] indicating that conventional automatic caption evaluation methods are biased and not well-suited for assessing detailed captions. Specifically, we utilize GPT-4o with the following prompt:
> ```
> You are a language expert evaluating the fluency of image captions.
>
> Fluency refers to how grammatically correct, natural, and well-formed the text sounds to a native English speaker. A fluent  caption should be grammatically correct, free of awkward phrasing, and read smoothly.
>
> Evaluate the fluency of the following caption and return your output **strictly in JSON format** with:
> - "reason": a key reason for your scoring
> - "score": a number between 0 (completely disfluent) and 100 (perfect fluency)
>
> Caption: "{caption}"
> ```
> In addition to evaluating the captions generated by CapMAS, we also assess human-written detailed captions for the same set of images.
>
> **Table A**
>
> |Captions generated by|Fluency &#8593;|
> |-|-|
> |Human|89.0|
> |CapMAS (LLaVA-v1.5-7B)|93.4|
> |CapMAS (LLaVA-NeXT-7B)|93.4|
> |CapMAS (LLaVA-NeXT-13B)|93.6|
> |CapMAS (InternVL-Chat-V1.5)|94.1|
>
> The results in Table A demonstrate that the captions generated by CapMAS achieve even higher fluency scores than the human-generated captions. This can be attributed to the final stage of CapMAS, in which the corrector LLM helps preserve or even improve the fluency of the captions.
>
> We sincerely thank the reviewer for suggesting this experiment, which enabled us to highlight a new strength of CapMAS in terms of fluency.
>
> >**Q3**. In Table 2, it appears that the CLAIR metric can reflect the introduced noises. It's not as problematic as the article suggests.
>
> **A3**. Yes, CLAIR is indeed capable of reflecting the introduced noise, which is precisely why we adopted it in our experiments. However, as shown in the prompt used below, the meaning of the score it produces is not clearly defined:
> ```
> On a precise scale from 0 to 100, how likely is it that the candidate caption is describing the same image as the reference  caption? (JSON format, with a key "score", value between 0 and 100, and a key "reason" with a string value.)
> ```
> In contrast, our proposed metrics have clearly defined interpretations and enable a more detailed analysis of MLLMs in terms of both factuality and coverage.
>
> [1] ImageInWords: Unlocking Hyper-Detailed Image Descriptions
> [2] Benchmarking and Improving Detail Image Caption
>
> **We have additional experimental results that we were unable to include here. To help us share them, we kindly ask you to click “Rebuttal Comment” to allow us to leave a supplementary comment.**

---

> > ### Comment · Reviewer_quCw · 2025-04-08
> >
> > Please show additional experimental results.

---

> > > ### Author Response · Authors · 2025-04-08
> > >
> > > Thank you for your comment! In this additional response, we demonstrate that inference-time scaling can improve the factuality–coverage trade-off, while alternative inference-time scaling methods are ineffective. We also provide experimental results for CapMAS on an additional dataset.
> > >
> > > # Improved Factuality–Coverage Trade-Off via Inference-Time Scaling #
> > > As we previously noted, CapMAS is designed to remove detected hallucinations rather than correct them. This design choice stems from our observation that attempts at correction often introduce new hallucinations. Consequently, applying CapMAS may eliminate not only hallucinations but also factual content, inherently leading to a trade-off between factuality and coverage.
> > >
> > > However, we show that this trade-off can be improved through inference-time scaling. Specifically, we demonstrate that applying CapMAS individually to multiple captions for a single image and then summarizing the results can lead to an overall enhancement in the caption quality for that image.
> > >
> > > Table B: Results of LLaVA-Next on a subset of IIW400
> > >
> > > |Number of CapMAS Applications|CLAIR|Factuality|Coverage|
> > > |-|-|-|-|
> > > |1|59.8|83.0|24.3|
> > > |2|63.3|76.8|32.6|
> > > |3|67.4|80.2|37.7|
> > > |4|68.2|79.9|38.9|
> > > |5|70.6|80.1|41.4|
> > >
> > > Table B demonstrates that the facutality-coverage trade-off and overall quality of the resulting captions improve as the number of CapMAS applications increases.
> > >
> > > # Effectiveness of the Self-Refine Method #
> > > We tested the effectiveness of a representative inference-time scaling approach, Self-Refine (SR) [3]. Since validating and revising one's output requires advanced reasoning capabilities, we conducted the experiments using GPT-4V. For SR, we used the following prompt:
> > > ```
> > > You are given an image and its corresponding caption.
> > > Your task is to:
> > > 1. Analyze the image and compare it with the caption.
> > > 2. Identify and correct any factual or descriptive errors in the caption based on the image.
> > > 3. Refine the caption for clarity, correctness, and completeness — even if the original caption is mostly accurate.
> > >
> > > Show your reasoning and then provide a final refined caption.
> > > ```
> > > <Table C>
> > >
> > > |Method|CLAIR|Factuality|Coverage|
> > > |-|-|-|-|
> > > |Base|82.4|77.1|53.5|
> > > |+SR (x1)|79.9|72.3|50.4|
> > > |+SR (x2)|79.1|70.6|50.1|
> > > |+SR (x3)|78.5|69.8|49.7|
> > >
> > > Table C shows that having the model revise its own captions iteratively does not lead to better results. This demonstrates the superiority of CapMAS in terms of inference-time scaling.
> > >
> > > [3] SELF-REFINE: Iterative Refinement with Self-Feedback
> > >
> > > # Evaluation of CapMAS on an Additional Dataset #
> > > To demonstrate the generalizability of CapMAS, we additionally tested its effectiveness on a subset of DOCCI (400 samples).
> > >
> > > <Table D>
> > >
> > > |Method|CLAIR|Factuality|Coverage|
> > > |-|-|-|-|
> > > |LLaVA-NeXT-7b|68.0|56.2|54.0|
> > > | +CapMAS|72.8|68.1|53.5|
> > > |LLaVA-NeXT-13b|70.1|58.2|55.6|
> > > | +CapMAS|74.5|74.6|52.3|
> > >
> > > Table D demonstrates that CapMAS is also effective on DOCCI.
> > >
> > > We sincerely appreciate your thoughtful feedback and have done our best to address your concerns. If there are any remaining or additional questions, please don’t hesitate to let us know. If you find our response satisfactory, we would be grateful if you could consider reflecting that in your score.
> > >
> > > Thank you again for your time and consideration.

---

### Decision · Program_Chairs · 2025-05-01

**Decision:**

Accept (poster)

**Comment:**

This paper addresses the challenge of hallucinations in hyper-detailed image captions generated by Multimodal Large Language Models (MLLMs). The authors posit that existing hallucination detection methods falter with long captions because MLLMs increasingly rely on their own generated text rather than the input image as sequences grow. To mitigate this, they propose CapMAS (Caption Multi-Agent System), a training-free, multi-agent approach where an LLM decomposes a given detailed caption into atomic propositions, and an MLLM verifies the factuality of each proposition against the image. Based on this verification (and a tunable threshold), a corrector LLM removes the non-factual propositions to produce a revised caption. Alongside the method, the paper introduces an evaluation framework comprising two new metrics, Factuality and Coverage, calculated using a similar decomposition-and-verification pipeline leveraging GPT-4o. They also curated and plan to release a VQA benchmark dataset to support this evaluation.

Reviewers largely agreed that the paper tackles an important and timely problem: reducing hallucinations in increasingly detailed image captions generated by MLLMs. They found the core CapMAS approach, which uses LLM-MLLM collaboration for decomposition, verification, and correction, to be logical and empirically effective at improving caption factuality across various base models, including strong ones like GPT-4V. Reviewers appreciated the training-free nature of the method and the introduction of new evaluation metrics (Factuality and Coverage) designed specifically for detailed captions, noting that the Factuality metric showed strong correlation with human judgments.

Reviewers identified several weaknesses and areas needing clarification. They questioned the precise contribution and necessity of the Coverage metric, observing that the CapMAS method often improves factuality at the expense of coverage due to its strategy of removing, rather than rewriting, hallucinated content – an inherent trade-off the authors acknowledged. Some reviewers also felt the evaluation datasets (IIW-400, DOCCI subset) were relatively small and suggested exploring broader benchmarks (for example DCI or PixelProse). Furthermore, reviewers noted similarities between CapMAS components and the proposed metrics to concepts in prior work (e.g., Woodpecker, ALOHa, POPE, Self-Refine), suggesting the novelty lies more in the effective combination and application to hyper-detailed captioning rather than entirely new techniques.

Overall, I recommend acceptance. But please take into account the reviewer comments seriously. Especially:
- better comparison with prior work
- inclusion of the rebuttal experiments into the main paper (datasets, calculations, fluency, qualitative analysis)
- better justification of the key insights and the important parts of the multi-step system
- more explanations of the cost and scalability of the CapMAS approach vs other options